# Using GDACS to anticipate clinical and operational burden after earthquakes: A global event-level analysis (2020–2024)

**Ahmet Aykut**◉*, **Ertuğ Günsoy, Cem Yıldırım**

Emergency Department, Health Sciences University (SBU), Van Education and Research Hospital, Van, Türkiye

* ahmet.aykut@gmail.com

## Abstract

Global Disaster Alert and Coordination System (GDACS) alerts are widely used after earthquakes, yet their clinical relevance is uncertain. We performed a retrospective, global, event-level study spanning 2020–2024. To avoid double counting, alerts were clustered into country-bounded representatives using a 48-hour gap, retaining the alert with the highest GDACS score (n = 85; Red = 17, Orange = 68). Primary outcomes were reported deaths and field-hospital deployment. Associations used Spearman correlation; deployment was modeled with Firth logistic regression. Sensitivity analyses used alternative deployment definitions and composite windows. The GDACS score correlated with deaths ($\rho = 0.522$, $p = 3.0 \times 10^{-7}$). Field-hospital deployment occurred in 52.9% of Red events and 0% of Orange events. The GDACS score strongly predicted deployment (OR=42.7, 95% CI 4.7–385.7), with AUC = 0.98 and Brier = 0.034. An exploratory exposure-normalized subset where GDACS reported population "within 100 km" (n = 19) showed directionally consistent results ($\rho = 0.50$, $p = 0.029$).GDACS metrics provide early, scalable indicators for surge planning, but are hazard- and exposure-centric and cannot capture mediators such as collapse dynamics or health-system resilience. Treating GDACS as a first-layer signal, complemented by subnational exposure and rapid damage assessment, can support more timely, evidence-based medical response after major earthquakes.

## Introduction

The Global Disaster Alert and Coordination System (GDACS) is a real-time information platform developed through a United Nations–European Commission partnership to support rapid situational awareness and coordination in major sudden-onset disasters. For earthquakes, GDACS combines event parameters (magnitude, depth, location) with population exposure, vulnerability, and country-level coping capacity to generate a continuous alert score and color-coded levels (Orange/Red) intended to

**Data availability statement:** "The dataset (85 composite representatives) and analysis scripts used in this study are publicly available on Zenodo at https://doi.org/10.5281/zenodo.17036717."

**Funding:** The author(s) received no specific funding for this work.

**Competing interests:** NO authors have competing interests.

flag events that may exceed national response capacity [1–3]. While these hazard- and exposure-centric alerts are widely used to trigger situational monitoring and coordination of Emergency Medical Teams (EMTs), the system's ability to anticipate clinical burden and operational health needs has not been systematically tested across diverse contexts [1,4].

Clinical impact after earthquakes is shaped not only by seismic energy but also by structural collapse dynamics, entrapment, access to care, and population factors. Building damage and entrapment drive trauma deaths and injury patterns, while secondary hazards and living conditions modulate downstream public-health effects [5,6]. In parallel, crush syndrome and acute kidney injury (AKI) are recurrent complications among earthquake survivors, with contemporary reports from the 2023 Türkiye earthquakes describing high AKI incidence and frequent dialysis requirements in multicenter and single-center cohorts, and recent reviews summarizing typical ranges and clinical mechanisms [7–9].

GDACS currently moderates alert scores using country-level indices of coping capacity, drawing on INFORM metrics; however, INFORM is designed for country-scale risk/severity situational awareness and may not capture subnational health-system fragility pertinent to near-term clinical surge [2,10–12]. Empirical evaluation is therefore needed to determine whether GDACS alert metrics align with observed mortality and medical deployment, and how demographic exposure and national capacity modify these associations. To address this gap, we analyze 85 composite earthquake events (2020–2024)—constructed to avoid double-counting within seismic sequences (composite definition in Methods)—to test whether higher GDACS scores correspond to greater mortality and field hospital deployment, and whether these relationships vary by region and exposure. As a co-primary endpoint, we also examine exposure-normalized mortality (deaths per 100,000 exposed) to improve comparability across differently sized populations.

## Materials and methods

### Study design and objectives

This retrospective, global, event-level observational study uses the composite earthquake event as the unit of analysis to avoid double-counting within seismic sequences. Within each country, earthquakes occurring within ±24 hours and ≤150 km (epicentral distance) were merged into a single composite; when coordinates were unavailable, same-day events within the same country were merged. For each composite we set deaths = sum, field-hospital deployment = max (any component event = 1), GDACS score = max, magnitude (Mw) = max, exposure within 100 km = max (to avoid double counting), and depth = median; the alert level was the highest observed (Red > Orange).

The study window was 1 January 2020–31 December 2024. After applying the composite rules and inclusion criteria, the final analytic sample comprised 85 composite events (Red = 17; Orange = 68).

**Primary objective.** To quantify the associational relationship between the GDACS composite score and (i) event-level mortality (deaths) and (ii) deployment of field

hospitals as an operational burden proxy; and to assess predictive performance for deployment (discrimination by AUC, overall accuracy by Brier score, and calibration slope/intercept).

**Secondary objectives.** (a) To examine exposure-normalized mortality (deaths per 100,000 exposed) to enhance comparability across differently sized populations; (b) to explore heterogeneity by continent and alert level (Red vs Orange); and (c) to conduct sensitivity analyses using the non-composite (raw) event list, an alternative deployment definition ("any temporary medical facility"), and alternative composite windows (±12/±48 hours; 100–200 km).

Given the observational design and event-level aggregation, all findings are associational rather than causal. Reporting follows relevant STROBE items for observational studies.

## Statistical analysis

All analyses were conducted at the composite event level. Continuous variables are summarized as median (IQR) and categorical variables as counts (percent). Between-group comparisons (Red vs Orange) used Mann–Whitney U tests; we additionally report Cliff's delta with bootstrap 95% CIs (2,000 resamples) as the effect size. Count distributions (deaths) were visualized on a log scale for readability while all inferential tests were performed on the original scale.

For correlational analyses, we estimated Spearman's $\rho$ between the GDACS composite score and two outcomes: (i) event-level deaths and (ii) exposure-normalized mortality, defined as

$$\text{deaths per } 100,000 \text{ exposed} = \frac{\text{deaths}}{\text{exposed within } 100 \text{ km}} \times 100000$$

excluding composites with zero or missing exposure from the rate analysis. We report two-sided p-values and control the false discovery rate (FDR) using Benjamini–Hochberg across each family of related tests (overall and continent-stratified correlations). Statistical significance was set at q < 0.05.

We computed Spearman rank correlations between the GDACS score and reported earthquake-related deaths within each continent. Two-sided p-values were obtained from scipy.stats.spearmanr. To control the false discovery rate across continent-level tests with n ≥ 3 composite events, we applied the Benjamini–Hochberg procedure and report q-values. Country names were standardized and continent labels assigned algorithmically; events spanning multiple continents were labeled "Transcontinental." Very small strata (e.g., n < 3) were treated as exploratory and not over-interpreted.

Our primary regression analysis modeled field-hospital deployment (yes/no) using logistic regression with the GDACS composite score as the sole predictor. Because class imbalance and near-/quasi-separation were anticipated (for example, zero deployments among Orange composite events), models were fitted using standard maximum-likelihood estimation, and ridge-penalized (L2) logistic regression was used as a robustness check to stabilize coefficient estimates. Odds ratios (ORs) are reported per 1-point increase in GDACS score with 95% confidence intervals. Model performance was evaluated by discrimination (AUC), overall accuracy (Brier score), and calibration (intercept and slope from regressing outcomes on predicted logits). Firth bias-reduced logistic regression was additionally explored in supplementary analyses and yielded qualitatively similar effect estimates (not shown in the main tables).

To account for potential overdispersion in death counts, we performed an exploratory sensitivity using negative binomial regression with an offset = log(exposed), estimating the association between GDACS and deaths on a rate scale. Results are presented as incidence-rate ratios (IRRs) with 95% CIs.

Missing data were handled by complete-case analysis for each model; variable-wise missingness is reported in the Supplement. As sensitivity analyses, we: (i) repeated the full pipeline on the non-composite (raw) event list; (ii) replaced the strict field-hospital outcome with a broader "any temporary medical facility" indicator; and (iii) varied the composite window (±12/±48 hours; 100–200 km) to test the stability of effect directions and magnitudes.

Analyses were implemented in R (for Firth logistic and calibration) and Python (for data processing, correlations, effect sizes, and penalized models); all code and data required to reproduce the tables and figures are openly archived. All tests were two-sided.

### Event selection and de-duplication

We considered all GDACS Orange or Red earthquake alerts issued between 1 January 2020 and 31 December 2024. From this universe we excluded non-earthquake hazards and entries with missing core identifiers (date, country, alert level). To avoid double-counting within multi-shock sequences, we constructed country-bounded composite representatives as follows: within each country, alerts were clustered using a 48-hour time-gap threshold; within each cluster, we selected the alert with the highest GDACS continuous score as the representative event. The representative events were flagged in the dataset (Use_Composite85 = 1). After this de-duplication, 85 representative events remained (Red = 17; Orange = 68) out of 95 raw alerts. Sequences spanning borders were not merged across countries; if the epicenter shifted across borders during a sequence, representatives were formed per country.

### Variables and linkage

All covariates and outcomes used in the analysis were derived from the GDACS alert feed and its hazard/exposure metadata, including the continuous GDACS score (0–10), alert level (Orange/Red), moment magnitude (Mw), hypocenter depth (km), and GDACS-modeled population exposed within 100 km, and were linked to the representative event (Table 1). Event-level deaths were abstracted from WHO/OCHA and national situation reports, consolidated using a fixed priority order (WHO/OCHA > national authorities > consolidated media) and linked to the representative event.

### Exposure-normalized mortality

As a co-primary endpoint, we computed deaths per 100,000 exposed. This rate was computed only for representatives whose GDACS record explicitly contained the string "within 100 km" in the exposure field; representatives with zero or missing exposure were excluded from this rate analysis. The resulting exposure-normalized subset comprised n = 19 events (Red n = 1; Orange n = 18). All other analyses used the full representative set (n = 85).

### Analytical note

All analyses were performed at the representative-event level; outcomes were defined as cumulative field-hospital deployment (yes/no), total reported deaths, and, in a subset, deaths per 100,000 exposed for each composite event. We did not model daily trajectories or lagged effects, because all outcome data were available only as event-level aggregates. Sensitivity analyses using alternative composite windows and outcome definitions are reported in the Supplement.

### Analytical environment and reproducibility

All workflows were scripted and executed from version-controlled notebooks. Data processing and non-Firth analyses used Python (pandas, numpy, scipy, statsmodels, scikit-learn); Firth logistic regression and calibration summaries used R (logistf, pROC, rms). Random seeds were fixed where applicable. All tables and figures are generated programmatically from the composite dataset to prevent transcription errors; the exact code used to create each table is archived with the dataset. Package versions and session information are provided in the Supplement to facilitate exact replication.

### Methodological considerations (bias and assumptions)

We address potential sources of bias as follows.

**Selection and case definition.** GDACS defines the alert universe; we restricted the study sample to Orange and Red alerts, following GDACS' own categorization of potentially damaging events and specifying this choice a priori.

**Table 1. GDACS indicators and operational definitions used in this study.**

| Indicator/ Variable | GDACS Source/ Origin | Units/ Scale | How Used In This Study | Thresholds and Rationale |
|---|---|---|---|---|
| GDACS continuous score | Core GDACS alert field | 0–10 (dimension-less index) | Continuous predictor of event severity in descriptive statistics, correlation analyses, and regression models. | No additional cut-off; modelled per 1-point increase. Standard GDACS composite indicator of hazard, exposure, and vulnerability; not previously validated for clinical outcomes, therefore treated as a continuous signal. |
| Alert level (color) | Core GDACS alert level | Categorical (Orange/ Red) | Defined the study universe; only Orange and Red alerts were eligible for inclusion. | Inclusion threshold = Orange or Red. Follows GDACS' own categorization of potentially damaging events; pre-specified a priori, not data-driven. |
| Moment magnitude (Mw) | GDACS hazard information | Continuous (Richter-equivalent) | Hazard-side covariate in descriptive tables and multivariable models. | No cut-off; analysed as a continuous descriptor (per 0.1 or 1-unit increase, depending on model specification). No established clinical threshold; used structurally. |
| Hypocentre depth | GDACS hazard information | Kilometres | Descriptive and exploratory covariate to characterise event geometry and potential shaking at the surface. | No cut-off; modelled per 10-km increase. Deeper events may attenuate surface damage; no validated depth threshold for clinical outcomes, so used exploratorily. |
| Population exposed within 100 km | GDACS modelled exposure field ("within 100 km") | Persons | Used when the exposure field explicitly referred to "population exposed within 100 km" to derive deaths per 100,000 exposed and to define the exposure-normalised mortality subset. | Analyses restricted to events with explicit "within 100 km" reporting. 100-km buffer is defined by GDACS; small n and partial exposure coverage are acknowledged as limitations. |
| Composite-event clustering rule | Derived from GDACS alerts using pre-specified windows | – | Within each country, alerts were clustered using temporal (±24 h) and spatial (≤150 km) windows; the alert with the highest GDACS score was selected as the representative composite event. | Temporal window ±24 h and spatial window ≤150 km were pre-specified pragmatic choices to avoid double-counting multi-shock sequences. Not previously validated for clinical outcomes; robustness assessed by varying temporal (±12–48 h) and spatial (100–200 km) windows (S1 Table). |
| Exposure-normalised mortality | Derived from GDACS exposure and documented deaths | Deaths per 100,000 exposed | Co-primary endpoint in the subset of events with explicit "within 100 km" exposure values. | Defined only for the subset with available exposure (n = 19 events). Provides a scale-free severity measure; interpretation remains cautious due to small sample size and incomplete exposure reporting. |

GDACS = Global Disaster Alert and Coordination System. Indicators are defined at the composite earthquake–event level and were extracted from the representative alert for each country-bounded event (clusters within ±24 hours and ≤150 km). The table lists each GDACS indicator used in the analysis, its source field, units and scale, and how it was operationalised, including any prespecified thresholds; alternative temporal and spatial windows for composite events are evaluated in sensitivity analyses (S1 Table).

Non-earthquake hazards and records with missing core identifiers were excluded. The composite event procedure (±24 h; ≤ 150 km within country) was pre-specified to avoid double-counting multi-shock sequences and represents a pragmatic temporal and spatial window informed by common seismic clustering practice rather than by prior validation for predicting clinical outcomes. Robustness of this choice is further assessed through alternative temporal and spatial windows in pre-specified sensitivity analyses (S1 Table).

**Outcome misclassification.** The primary deployment outcome uses a strict field-hospital definition; a broader "any temporary medical facility" indicator is reported as a sensitivity analysis. Deaths were consolidated using a fixed priority order (WHO/OCHA > national authorities > consolidated media).

**Confounding and scale.** Analyses are event-level and associational; residual confounding by unmeasured subnational fragility or access cannot be ruled out. To partially address this, we report exposure-normalized mortality and examine alternative outcome definitions in sensitivity analyses; however, we were unable to fit stable multivariable deployment models because exposure within 100 km was available only for a small subset of events, so residual confounding cannot be excluded.

**Multiple testing.** We control the false discovery rate (Benjamini–Hochberg) within families of related tests; $q < 0.05$ denotes statistical significance.

**Missing data.** Complete-case analyses were used per model; variable-wise missingness is tabulated in the Supplement.

**Robustness.** We conducted pre-specified sensitivity analyses: (i) raw (non-composite) events; (ii) alternative deployment definition; and (iii) alternative composite windows. Directions and magnitudes of the main associations were monitored for stability, and the specific alternative temporal and spatial windows are detailed in S1 Table.

## Ethical considerations

The study used publicly available, event-level aggregated information from international humanitarian and governmental portals without any individual-level identifiers or interventions. In accordance with institutional policies for research using non–human-subjects data, the project qualifies for exemption from human subjects review. All procedures are consistent with the principles of the Declaration of Helsinki, and no patient consent was required.

## Results

Eighty-five composite earthquake events occurring between 1 January 2020 and 31 December 2024 met inclusion criteria after de-duplication of multi-shock sequences. Events spanned Asia, the Americas, and Oceania and included both Orange and Red alerts (Red = 17; Orange = 68). Summary statistics for seismological parameters, GDACS metrics, and the modeled exposed population within 100 km are presented in Table 2.

The GDACS continuous score showed a statistically significant, moderate positive correlation with reported deaths (Spearman's $\rho = 0.520$, $p = 3.43 \times 10^{-7}$; BH-FDR $q = 6.87 \times 10^{-7}$; $n = 85$). As visualized in Fig 1, higher GDACS scores were generally associated with higher mortality; the y-axis is log-scaled for presentation, whereas all inference used raw counts. Mortality was substantially higher among Red versus Orange alerts; medians and IQRs are reported in Table 2 (non-parametric contrasts confirm this difference; see legend of Fig 2A).

Operational indicators paralleled these clinical patterns. Using the strict field-hospital definition, deployment occurred in 52.9% of Red events (9/17) and in 0% of Orange events (0/68); Fisher's exact test supported a significant difference. In logistic regression, the GDACS score was a strong independent predictor of field-hospital deployment: odds ratio

**Table 2. Descriptive characteristics of representative (composite) earthquake events (2020–2024).**

| Group | Overall | Orange | Red |
|---|---|---|---|
| N | 85 | 68 | 17 |
| Deaths (median) | 1 | 0.5 | 157 |
| Deaths (Q1) | 0 | 0 | 3 |
| Deaths (Q3) | 8 | 3 | 1482 |
| Exposed 100 km (median) | $2.82 \times 10^6$ | $2.70 \times 10^6$ | $3.80 \times 10^6$ |
| Exposed 100 km (Q1) | $1.30 \times 10^6$ | $1.28 \times 10^6$ | $3.80 \times 10^6$ |
| Exposed 100 km (Q3) | $3.77 \times 10^6$ | $3.64 \times 10^6$ | $3.80 \times 10^6$ |
| Deaths/100k (median) | 0 | 0 | 0.079 |
| Deaths/100k (Q1) | 0 | 0 | 0.079 |
| Deaths/100k (Q3) | 0.054 | 0.033 | 0.079 |
| Field-hospital (n) | 9 | 0 | 9 |
| Field-hospital (rate) | 10.6% | 0.0% | 52.9% |

Values are median (IQR) or n (%). 'Deaths/100k' excludes representatives with zero or missing exposure; exposure is the GDACS-modeled population within 100 km (when explicitly reported). Composite representatives were constructed by clustering alerts within a country using a 48-hour time-gap and selecting the alert with the highest GDACS score. 'Deaths/100k' excludes representatives with zero or missing exposure; exposure is the GDACS-modeled population within 100 km (when explicitly reported).

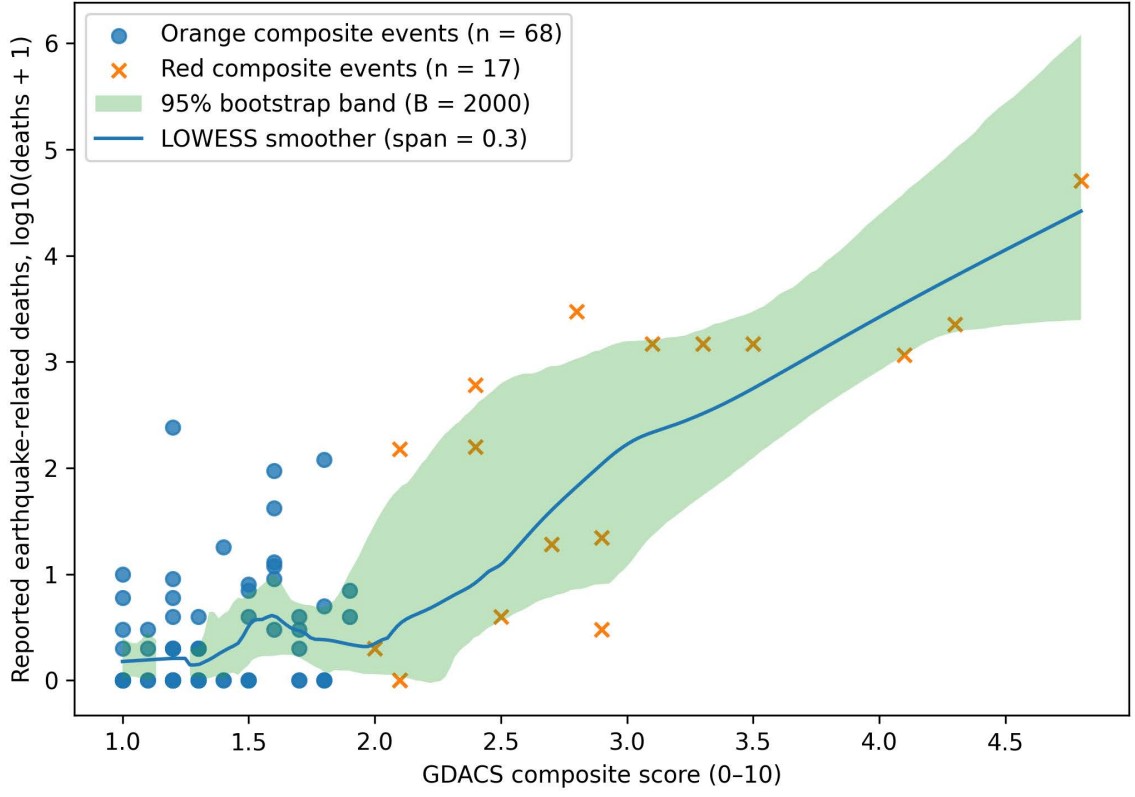

**Fig 1. Association between GDACS composite score and reported earthquake-related deaths (2020–2024).** The x-axis shows the GDACS continuous composite score (range 0–10; higher values indicate greater combined hazard and population exposure). The y-axis shows reported earthquake-related deaths on a $\log_{10}$ scale, plotted as $\log_{10}(\text{deaths}+1)$ to display events with zero deaths. Each marker represents one composite earthquake event (N = 85); orange circles indicate Orange alerts (n = 68) and blue crosses indicate Red alerts (n = 17). The solid line shows a locally weighted scatterplot smoother (LOWESS, span = 0.3) with a 95% bootstrap confidence band (B = 2000). Spearman's rank correlation between the GDACS score and deaths was $\rho = 0.520$ ($p = 3.43 \times 10^{-7}$; $q = 6.87 \times 10^{-7}$). All statistical inference used raw (untransformed) death counts.

(OR) = 42.72 (95% CI 4.73–385.71), with AUC = 0.981 and Brier = 0.034 (Table 3). A ridge-penalized sensitivity analysis yielded a smaller but directionally consistent effect (OR ≈ 9.69) with comparable discrimination (AUC ≈ 0.98), indicating robustness to near-separation.

To enhance comparability across differently sized populations, we additionally examined mortality per 100,000 exposed in a limited subset (n = 19) where GDACS explicitly reported exposure within 100 km. Findings were directionally consistent with the count outcome; however, given the small n (only one Red event), this analysis is exploratory and should not be interpreted as confirmatory. Fig 2B illustrates these exploratory distributions.

In continent-stratified analyses of composite representatives, Asia (n = 54) showed a strong positive correlation between GDACS score and reported deaths (Spearman's $\rho = 0.57$; $p < 0.001$; $q < 0.001$). A similar positive association was observed in South America (n = 8; $\rho = 0.79$; $p = 0.0196$; $q = 0.039$). North America (n = 8) showed a concordant direction of effect with greater uncertainty ($\rho = 0.63$; $p = 0.094$; $q = 0.126$). Oceania (n = 10) showed only a weak, non-significant association ($\rho = 0.27$; $p = 0.443$; $q = 0.443$). Europe (n = 2), Africa (n = 2), and the single transcontinental composite were too sparse for inferential interpretation and are reported descriptively only (Table 4).

Effective sample sizes varied slightly across analyses due to occasional gaps in operational reporting and exclusion of composites with zero/missing exposure from rate calculations; complete-case analysis was used throughout. Denominators for each test are specified in the corresponding tables and figure legends.

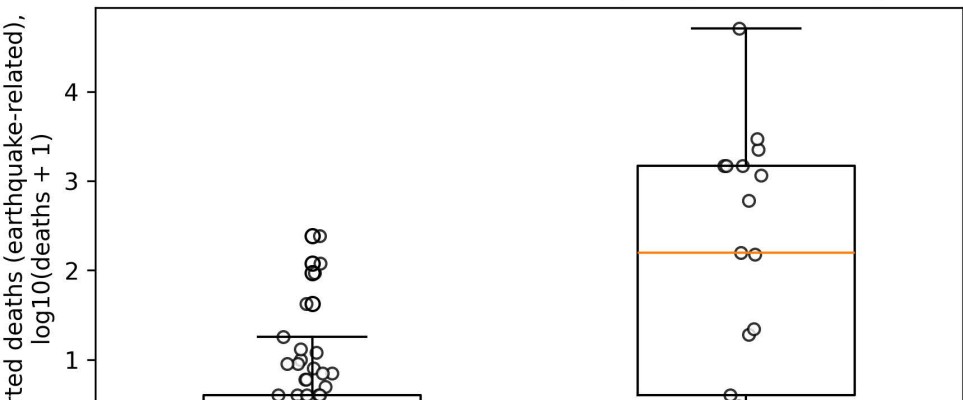

### A. Deaths by GDACS alert level (composite events)

n(Orange) = 68, n(Red) = 17; U = 973, p < 0.001, Cliff's Δ = 0.68

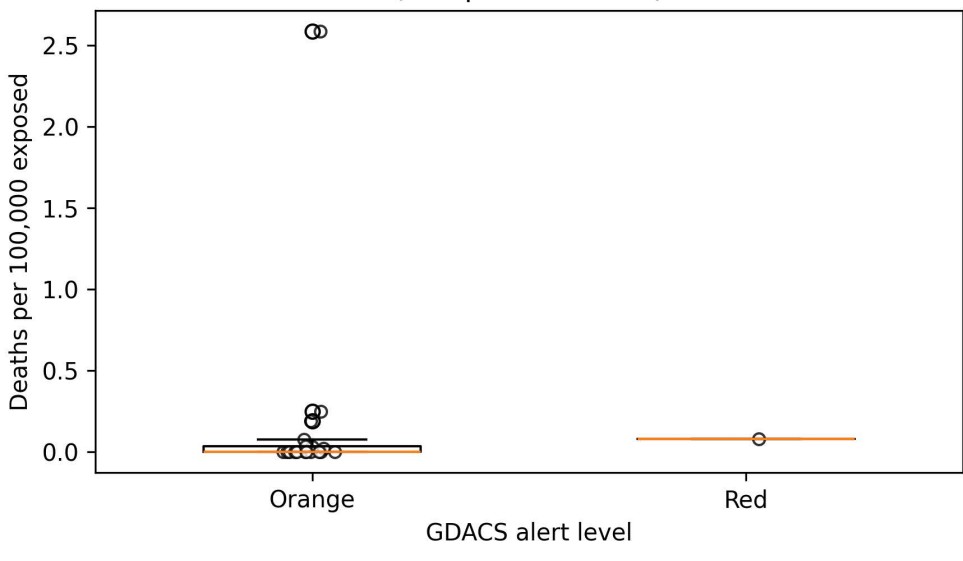

### B. Exposure-normalized mortality by GDACS alert level (composite events)

n(Orange) = 18, n(Red) = 1; U = 15, p = 0.264, Cliff's Δ = 0.67

**Fig 2. Mortality by GDACS alert level with and without exposure normalization (composite earthquake events, 2020–2024).** Orange = 68 composite events, Red = 17. A. Event-level deaths. Boxplots show reported earthquake-related deaths by alert level. The y-axis uses a $\log_{10}(\text{deaths} + 1)$ transformation for visualization; all statistical inference used raw death counts. Mann–Whitney U (Red vs Orange) = 973, p = $6.36 \times 10^{-6}$; Cliff's Δ = 0.68 (large effect). B. Exposure-normalized mortality. Deaths per 100,000 exposed were computed as deaths/(GDACS-modeled population exposed within 100 km) × 100,000 and are shown only for composite events where GDACS explicitly reported exposure "within 100 km". The resulting subset comprised n = 19 events (Red n = 1, Orange n = 18). Given the small sample and single Red event, this comparison is considered exploratory.

**Table 3. Logistic models predicting field-hospital deployment from GDACS score (composite events, 2020–2024).**

| Model | Model A (GDACS only) | Model A (GDACS only) |
|---|---|---|
| **Estimator** | Standard logistic (ML) | Ridge logistic (L2) |
| **N** | 85 | 85 |
| **OR per 1-point GDACS (95% CI)** | 42.72 (4.73–385.71) | 9.69 (–)* |
| **AUC** | 0.981 | 0.982 |
| **Brier** | 0.034 | 0.034 |
| **Calibration slope** | – | 1,65 |
| **Calibration intercept** | – | 0,91 |

Model A: deployment (field hospital, strict definition) ~ GDACS composite score (per 1-point increase). Estimators shown are standard maximum-likelihood logistic and ridge (L2-penalized) logistic regression. Odds ratios (ORs) are per 1-point increase in GDACS score. Discrimination is summarised by the area under the ROC curve (AUC), overall accuracy by the Brier score, and calibration by slope and intercept where available. All models use the full composite dataset (N = 85).

**Table 4. Continent-stratified association between GDACS score and reported deaths.**

| Continent | N | Spearman's ρ | p value | q value (BH) |
|---|---|---|---|---|
| Europe | 2 | 1.000 | – | – |
| Asia | 54 | 0.570 | <0.001 | 0.001 |
| Africa | 2 | 1.000 | – | – |
| North America | 8 | 0.629 | 0.094 | 0.126 |
| South America | 8 | 0.790 | 0.020 | 0.039 |
| Oceania | 10 | 0.274 | 0.443 | 0.443 |
| Transcontinental | 1 | – | – | – |

p-values are reported to three decimals; values <0.001 are shown as "<0.001". q-values are Benjamini–Hochberg–adjusted within the continent-level family and reported to three decimals (minimum 0.001). Strata with n < 3 are descriptive only and not interpreted inferentially.

## Discussion

This global evaluation indicates that GDACS alert metrics are associated with both clinical and operational burden following major earthquakes. Across 85 composite events (2020–2024), higher GDACS scores were moderately correlated with reported mortality and were strongly linked to field-hospital deployment, while Red alerts showed markedly higher deaths and operational activation than Orange alerts. These findings support using GDACS outputs as early, scalable indicators for surge planning, rather than as stand-alone clinical predictors [1–4].

The continent-level pattern largely mirrors the global signal: the GDACS score tends to increase with observed mortality, with the strongest evidence in Asia and supportive findings in South America. Regions with small samples (Europe, parts of the Americas) exhibit wide uncertainty, cautioning against over-interpretation of rank correlations in sparse strata. Collectively, these results suggest that the GDACS composite captures aspects of event severity relevant to clinical impact, while highlighting regional heterogeneity and the importance of data density for stable estimation.

Two features strengthen the interpretation of these associations. First, the exposure-normalized mortality endpoint (deaths per 100,000 exposed) reproduced the direction and statistical significance of the count-based relationship with GDACS in the limited subset with exposure data (Spearman's ρ ≈ 0.50; q ≈ 0.03; Fig 2B and S1 Table), although this analysis remains exploratory given the small sample (n = 19, including only one Red event). Second, logistic models predicting field-hospital deployment from the GDACS score showed robust performance and remained stable across maximum-likelihood and ridge-penalized estimators (Table 4); discrimination (AUC) and overall accuracy (Brier) were favorable

[13,16,17]. Model-fit diagnostics are summarized in S2 Table, and results were qualitatively unchanged under alternative specifications in sensitivity analyses (S1 Table) [18].

At the same time, the magnitude of associations and their contextual variability highlight the limits of hazard-centric alerts for inferring clinical burden. Earthquake morbidity and mortality are shaped by structural collapse, entrapment duration, environmental exposure, and access to care—determinants that vary across settings and are not directly encoded in GDACS scoring [5,6]. Post-earthquake complication profiles such as crush syndrome and acute kidney injury (AKI)—prominent in recent cohorts from the 2023 Türkiye earthquakes—carry implications for dialysis and critical-care demand, yet these clinical mediators are absent from current global alert algorithms [7–9]. In line with this, demographic exposure aligns with mortality patterns, whereas a country-level coping index (e.g., INFORM) is not consistently associated at the event level, underscoring that national aggregates can miss subnational health-system fragility relevant to near-term clinical outcomes [10–12].

Operational findings are congruent with contemporary coordination practice: GDACS alerts and WHO–OCHA reporting often precede or accompany EMT mobilization, and we observed substantially higher deployment in Red alerts [1,4]. From a policy perspective, these results argue for treating GDACS as a first-layer signal that triggers situational monitoring and provisional contingency actions, while downstream decisions (e.g., dialysis capacity staging, critical-care referral pathways) should integrate rapidly evolving information on hospital functionality, lifelines, and anticipated case-mix.

Several robustness checks mitigate concerns that results hinge on a single extreme sequence or analytic choices. Associations persisted after excluding the 2023 Türkiye (Kahramanmaraş) sequence, and rate-normalized endpoints yielded concordant conclusions (S1 Table). Addressing quasi-separation with Firth bias-reduced logistic regression in additional sensitivity analyses produced effect sizes concordant with maximum-likelihood estimates and preserved discrimination, suggesting that the observed operational signal is not an artifact of sparse outcomes [13,14,16,17]. For exploratory correlation families (e.g., continent-level strata), nominal p-values were contextualized with false discovery rate control; these adjustments did not alter qualitative conclusions [15]. Nonetheless, between-country heterogeneity in how "field hospitals" and temporary facilities are reported—particularly distinctions between domestic vs international EMTs—remains an important source of classification variability to standardize in future work.

This study has practical implications and research directions. Near term, integrating GDACS hazard signals with subnational exposure maps, rapid damage assessments, and health-system status could better align alerts with expected clinical burden. Medium term, structured extraction of operational and clinical variables from narrative situation reports (e.g., standardized EMT typologies, dialysis demand, oxygen/blood product constraints) would improve comparability across countries and events. Longer term, incorporating complication-risk models (e.g., crush injury/AKI risk under varying building stock and rescue times) and hospital-resilience indicators into alert logic may narrow the gap between hazard detection and bedside realities [1–4,7–9].

In sum, GDACS functions as an effective global early-warning platform whose metrics show consistent associations with mortality and emergency medical deployment. Its clinical precision is necessarily limited by the absence of direct measures of infrastructure damage, care accessibility, and injury patterns. Interpreting GDACS outputs as first-layer indicators—augmented by exposure-normalized outcomes and standardized operational data—offers a pragmatic path to evidence-based surge planning in the immediate aftermath of major earthquakes.

## Limitations

This study has several limitations. First, all outcomes were abstracted from secondary humanitarian and governmental sources, which vary in reporting quality, timeliness, and operational definitions. Although consolidated figures near event onset were prioritized and multiple documents cross-checked where available, differential completeness and case ascertainment may have led to underestimation or misclassification of deaths and deployments. Second, operational outcomes—field-hospital deployment and the establishment of temporary medical facilities—were not uniformly standardized

across countries, and distinctions between domestic and international EMTs were not consistently reported. This hetero-geneity introduces classification error that could attenuate associations and complicate cross-country comparability [1,4].

Third, GDACS alerting is primarily hazard- and exposure-centric; it does not directly incorporate clinical mediators such as building-collapse rates, rescue times, injury profiles, or health-system functionality, all of which influence morbidity and mortality [1–3,5,6]. The INFORM Risk Index was considered as a proxy for national coping capacity, but its country-level scale may miss subnational variation pertinent to event-level clinical burden, which likely contributes to inconsistent asso-ciations at the event level [10–12]. Fourth, while we complemented death counts with exposure-normalized mortality, the exposed-population denominator was available for a subset of events (as reflected in Table 2), limiting statistical power for rate-based analyses.

Fifth, complete-case analyses were used without multiple imputation because missingness largely reflected absent out-come reports rather than partial covariates; residual bias from differential data completeness cannot be excluded. Sixth, logistic models addressed near-/quasi-separation with Firth bias-reduced estimation and demonstrated favorable discrim-ination and calibration; nevertheless, the number of deployments was modest, resulting in wide confidence intervals in some specifications, and model-fit metrics should be interpreted with appropriate caution [13–18]. Seventh, subgroup and stratified analyses—by continent and by high magnitude—were exploratory and constrained by small strata, raising type-I error concerns despite multiplicity-aware interpretation.

Finally, the composite-event design reduces double counting in multi-shock sequences but aggregates within-sequence heterogeneity (e.g., spatially heterogeneous damage across closely timed shocks). The study period (2020–2024) and the focus on Orange/Red alerts also limit generalizability to other time frames and lower-severity events.

## Conclusion

GDACS alert metrics show consistent associations with clinical and operational burden after major earthquakes. In an event-level analysis of 85 composite earthquake events (2020–2024), higher GDACS scores were moderately correlated with mortality and strongly associated with field-hospital deployment (strict definition); these relationships persisted when mortality was normalized by exposed population and when near-/quasi-separation in deployment was addressed with Firth bias-reduced logistic regression. Red alerts corresponded to markedly greater deaths and operational activation than Orange alerts. Taken together, these findings support the use of GDACS outputs as early, scalable indicators for surge planning, while underscoring that hazard metrics alone are insufficient to capture the full spectrum of clinical determinants [1–4].

Improving clinical precision will require fusing GDACS hazard signals with subnational exposure maps, rapid damage assessments, and health-system status, as well as standardized operational reporting that distinguishes domestic and international EMTs. Routine availability of exposure-normalized endpoints and structured extraction of clinical variables from narrative reports would enhance comparability across settings. In the longer term, integrating complication-risk models and hospital-resilience indicators into alert logic may narrow the gap between early warnings and bedside realities [5–12]. Until such integration is achieved, GDACS should be interpreted as a first-layer indicator that can trigger situa-tional monitoring and provisional contingency actions, complemented by context-specific data to guide evidence-based medical response.

## Supporting information

**S1 Table. Sensitivity analyses for primary associations.** Event-level sensitivity analyses for the association between GDACS indicators and outcomes across 85 composite earthquake events (2020–2024). Each row represents a pre-specified robustness check: exclusion of the February 2023 Türkiye sequence, use of exposure-normalized mortality (deaths per 100,000 exposed), comparison of deaths between Red and Orange alerts, and alternative deployment definitions in logistic models (any temporary facility vs field hospital only). Columns report the number of events (N), the

effect metric (Spearman's ρ, Mann–Whitney U, or odds ratio per 1-point increase in GDACS score), point estimates with 95% confidence intervals where applicable, p values, Benjamini–Hochberg–adjusted q values, and brief interpretive notes. GDACS = Global Disaster Alert and Coordination System; OR = odds ratio; CI = confidence interval; ML = maximum likelihood.
(DOCX)

**S2 Table. Model diagnostics and missingness.** Panel A summarises apparent discrimination and overall accuracy for maximum-likelihood logistic models predicting field-hospital deployment and any temporary facility deployment from the GDACS composite score (N = 85 composite events, 2020–2024). Reported statistics include the Hosmer–Lemeshow goodness-of-fit $\chi^2$ test (degrees of freedom, p value), the area under the receiver-operating characteristic curve (AUC), and the Brier score; calibration slope and intercept are derived from same-sample recalibration (apparent calibration). External validation was not performed. Panel B reports variable-wise missingness counts in the analytic dataset (N = 85), indicating complete availability of core GDACS and outcome variables and partial availability of GDACS-modelled exposure within 100 km. GDACS = Global Disaster Alert and Coordination System; AUC = area under the ROC curve.
(DOCX)

## Author contributions

**Conceptualization:** Ahmet Aykut.

**Data curation:** Cem Yıldırım, Ertuğ Günsoy.

**Formal analysis:** Ahmet Aykut, Cem Yıldırım.

**Methodology:** Ahmet Aykut.

**Project administration:** Ahmet Aykut.

**Supervision:** Ahmet Aykut.

**Validation:** Ertuğ Günsoy.

**Visualization:** Ahmet Aykut.

**Writing – original draft:** Ahmet Aykut.

**Writing – review & editing:** Cem Yıldırım, Ertuğ Günsoy.

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
