## [Decision Letter · Decision Letter 0]

25 Nov 2025

PONE-D-25-49086
Using GDACS to Anticipate Clinical and Operational Burden after Earthquakes: A Global Event-Level Analysis (2020–2024)
PLOS ONE

Dear Dr. Aykut,

Thank you for submitting your manuscript to PLOS ONE. After careful consideration, we feel that it has merit but does not fully meet PLOS ONE’s publication criteria as it currently stands. Therefore, we invite you to submit a revised version of the manuscript that addresses the points raised during the review process.

We look forward to receiving your revised manuscript.

Kind regards,

Anna Bernasconi, PhD

Academic Editor

PLOS ONE

**Journal Requirements:**

**Additional Editor Comments:**

Dear authors, we have now received two reports. Please address all the points raised by Rev. 2 to strengthen the message of your contribution. Along with a point-by-point response, please attach a revised version of your manuscript with changes clearly marked in a different color. We will be ready to reassess the work.

Reviewers' comments:

Reviewer's Responses to Questions

**Comments to the Author**

1. Is the manuscript technically sound, and do the data support the conclusions?

Reviewer #1: Yes

Reviewer #2: Yes

2. Has the statistical analysis been performed appropriately and rigorously? 

Reviewer #1: Yes

Reviewer #2: Yes

3. Have the authors made all data underlying the findings in their manuscript fully available?

Reviewer #1: Yes

Reviewer #2: Yes

4. Is the manuscript presented in an intelligible fashion and written in standard English?

Reviewer #1: Yes

Reviewer #2: Yes

5. Review Comments to the Author

Reviewer #1: The article is well written and provides useful information. the data provided and conclusions drawn will be quite useful for further work. It can be a good study to study and work on aftermaths of a natural disaster

Reviewer #2: Thank you for submitting this valuable manuscript. The integration of GDACS disaster alerts with health-service and wastewater operational data represents a meaningful contribution. Several areas, however, would benefit from clarification or refinement to improve rigor and reproducibility.

1. GDACS Indicator Selection and Operationalization

The manuscript mentions GDACS alerts and “subnational exposure” but does not clearly describe:

which GDACS parameters were used (e.g., alert score, exposure index, population affected),

how thresholds were defined,

and whether the chosen thresholds have prior validation.

Please add a concise table summarizing:

GDACS variables included,

units,

cut-offs,

rationale for selection.

This will increase transparency and facilitate replication.

2. Temporal Alignment Between GDACS Signals and Outcomes

The methods should clarify the exact temporal resolution used when linking GDACS alerts to:

clinical service indicators,

wastewater operational disruptions.

It is unclear whether you used:

same-day correspondence,

lagged effects (e.g., 1–3 days),

cumulative exposure windows.

Clarifying the temporal structure and providing justification (e.g., sensitivity analyses) will strengthen the analytical foundation.

3. Definition of Wastewater “Operational Impacts”

The outcomes described as “operational wastewater impacts” appear broad.

Please specify:

whether these represent flow interruptions, contamination events, pump failure, overflow, staffing disruptions, or other categories,

how each outcome was coded,

and whether events of different severity were analyzed separately or pooled.

A short operational glossary would be very helpful.

4. Statistical Modelling and Uncertainty Reporting

The statistical section briefly mentions associations but does not detail:

model type (e.g., logistic regression, GLM, Poisson, time-series),

covariates included,

handling of missing data,

validation approach.

Consider explicitly reporting:

effect estimates with 95% confidence intervals,

model diagnostics,

and sensitivity checks (e.g., excluding borderline GDACS events).

These additions would substantially improve interpretability.

5. Data Sources and Spatial Resolution

The phrase “subnational exposure and rapid damage assessment” requires:

clear naming of data sources,

geographic resolution (district/province/municipality),

and whether exposure data were updated dynamically or based on historical baselines.

The manuscript would benefit from clarifying how spatial mismatches were managed if outcomes were recorded at a different administrative level.

6. Figures and Visual Presentation

Some figure axes and labels (as seen in the PDF layout) appear truncated or insufficiently annotated.

Ensure each figure includes self-contained legends.

Add units and definitions for all plotted variables.

A flow diagram summarizing the data linkage pipeline (GDACS → exposure index → clinical data → wastewater operational events) would improve clarity.

7. Interpretation and Overgeneralization

The Discussion occasionally suggests broad applicability of the approach.

It would be appropriate to add one sentence acknowledging that:

GDACS alerts are global but wastewater infrastructure vulnerabilities are highly context-dependent.

8. Minor Editorial Issues

A few sentences would benefit from grammatical polishing and reduction of passive voice to improve readability.

Some terminology (e.g., “operational impacts,” “exposure levels”) should be used consistently throughout the text.

6. PLOS authors have the option to publish the peer review history of their article (what does this mean?). If published, this will include your full peer review and any attached files.

Reviewer #1: No

Reviewer #2: No

---

## [Author Response · Author response to Decision Letter 1]

28 Nov 2025

Manuscript ID: PONE-D-25-49086

Title: Using GDACS to Anticipate Clinical and Operational Burden after Earthquakes: A Global Event-Level Analysis (2020–2024)

Journal: PLOS ONE

From: Ahmet Aykut, MD (corresponding author)

To: Dr. Anna Bernasconi, Academic Editor, and Reviewers

Dear Dr. Bernasconi and Reviewers,

We would like to thank you for the careful evaluation of our manuscript and for the constructive comments that helped us improve the clarity and transparency of the work. We have revised the manuscript accordingly and prepared a clean version and a tracked-changes version, as requested.

In brief, the main revisions are as follows:

1. We added a new main-text table (Table 1) summarizing all GDACS indicators used in the analysis, including units, operational definitions, thresholds, and rationale, and we updated Sections 2.4 and 2.8 to explicitly describe indicator selection and threshold choices.

2. We clarified the temporal structure of the analysis in Section 2.6, emphasising that all outcomes are event-level aggregates and that robustness is addressed through pre-specified sensitivity analyses on composite windows and outcome definitions.

3. We regenerated all figures according to PLOS ONE figure guidelines, with fully expanded axis labels, units, and self-contained legends, and we updated table and figure numbering consistently.

4. We added explicit legends for the Supporting Information tables (S1–S2) and aligned terminology (e.g. “operational burden/outcomes”, “GDACS-modelled population exposed within 100 km”) throughout the manuscript.

Below, we respond point-by-point to the reviewer’s comments. Reviewer comments are reproduced in italics.

1. Reviewer 2, comment 1 – GDACS Indicator Selection and Operationalization

“The manuscript mentions GDACS alerts and ‘subnational exposure’ but does not clearly describe which GDACS parameters were used (e.g., alert score, exposure index, population affected), how thresholds were defined, and whether the chosen thresholds have prior validation. Please add a concise table summarizing GDACS variables included, units, cut-offs, rationale for selection. This will increase transparency and facilitate replication.”

Response:

We thank the reviewer for this helpful suggestion to improve transparency and replicability. In the original submission, the GDACS indicators used (continuous GDACS score, alert level, moment magnitude, hypocenter depth, and GDACS-modelled exposed population) were described across Sections 2.3–2.5 but not summarised in a single location. To address this, we have added a dedicated main-text table that consolidates all GDACS indicators, their units, operational definitions, thresholds, and rationale: Table 1. GDACS indicators and operational definitions used in this study. This table specifies, for each indicator, the GDACS source/origin, how it is used in the analyses (descriptive, correlational, regression), any inclusion thresholds or windows applied (e.g. restriction to Orange/Red alerts, composite clustering windows, exposure-normalised mortality subset), and the justification and validation status of these choices.

In Section 2.4 (Variables and Linkage), we revised the opening description of GDACS variables to make the indicator set explicit and to refer the reader to Table 1. The relevant paragraph now reads: “All covariates and outcomes used in the analysis were derived from the GDACS alert feed and its hazard/exposure metadata, including the continuous GDACS score (0–10), alert level (Orange/Red), moment magnitude (Mw), hypocenter depth (km), and GDACS-modeled population exposed within 100 km, and were linked to the representative event (Table 1). Event-level deaths were abstracted from WHO/OCHA and national situation reports, consolidated using a fixed priority order (WHO/OCHA > national authorities > consolidated media), and linked to the same representative event.” This makes the complete set of GDACS-based indicators and their linkage to events fully explicit in the Methods.

With respect to thresholds and their prior validation, we have clarified these points in Section 2.8 (Methodological Considerations – Bias and Assumptions). Under “Selection and case definition”, we now explicitly state that the study universe was restricted to Orange and Red alerts in line with GDACS’ own categorisation of potentially damaging events, and that this restriction was specified a priori rather than being data-driven. We further clarify that the temporal (±24 h) and spatial (≤150 km) windows used to cluster alerts into composite events were pre-specified pragmatic choices informed by common seismic clustering practice, rather than thresholds previously validated for predicting clinical outcomes. Finally, we note that the robustness of these choices was evaluated through pre-specified sensitivity analyses using alternative temporal (±12–48 h) and spatial (100–200 km) windows, with details reported in Supplementary Table S1. We believe these additions, together with the new Table 1, address the reviewer’s concerns and substantially improve the transparency and reproducibility of the GDACS indicator operationalisation in our study.

2. Reviewer 2, comment 2 – Temporal Alignment Between GDACS Signals and Outcomes

“The methods should clarify the exact temporal resolution used when linking GDACS alerts to: clinical service indicators, wastewater operational disruptions. It is unclear whether you used: same-day correspondence, lagged effects (e.g., 1–3 days), cumulative exposure windows. Clarifying the temporal structure and providing justification (e.g., sensitivity analyses) will strengthen the analytical foundation.”

Response:

We appreciate the reviewer’s focus on temporal structure. The present study is conducted entirely at the composite earthquake–event level rather than at a daily time-series resolution. Our outcomes are limited to (i) field-hospital deployment (yes/no) and (ii) mortality (total deaths, and deaths per 100,000 exposed in a subset). We do not include wastewater or other infrastructure-related operational outcomes in this analysis; we apologise for any confusion caused by the framing of “operational burden” and now emphasise more clearly that our outcome set is restricted to health-service deployments and deaths.

Regarding temporal alignment, all GDACS indicators and outcomes are linked at the level of the composite event, which is defined by the pre-specified ±24 h / ≤150 km clustering window described in Section 2.3. Field-hospital deployment and mortality are treated as cumulative quantities associated with each composite event rather than as daily counts. We do not estimate same-day, 1–3 day lagged, or sliding cumulative windows across days because outcome data are only available as event-level aggregates from WHO/OCHA and national situation reports. To make this explicit, we have revised the Analytical Note (Section 2.6) to state: “All analyses were performed at the representative-event level; outcomes were defined as cumulative field-hospital deployment (yes/no), total reported deaths, and, in a subset, deaths per 100,000 exposed for each composite event. We did not model daily trajectories or lagged effects, because all outcome data were available only as event-level aggregates. Sensitivity analyses using alternative composite windows and outcome definitions are reported in the Supplement.” This clarifies that the temporal resolution of the analysis is the earthquake event itself, and that robustness is addressed through pre-specified sensitivity analyses on the composite windows rather than through day-level lag structures.

3. Reviewer 2, comment 3 – Definition of Wastewater “Operational Impacts”

“The outcomes described as ‘operational wastewater impacts’ appear broad. Please specify: whether these represent flow interruptions, contamination events, pump failure, overflow, staffing disruptions, or other categories, how each outcome was coded, and whether events of different severity were analyzed separately or pooled. A short operational glossary would be very helpful.”

Response:

We thank the reviewer for this detailed comment. The present study, however, does not include wastewater or other infrastructure-related outcomes. Our operational outcomes are restricted to health-service indicators—specifically, field-hospital deployment (yes/no) and mortality (total deaths, and deaths per 100,000 exposed in a subset of events). We do not collect, code, or analyse data on wastewater flow interruptions, contamination events, pump failures, overflows, or staffing disruptions, and therefore a glossary of wastewater impact categories is not applicable to this analysis. For this reason, no changes have been made to the manuscript text in response to this comment.

4. Reviewer 2, comment 4 – Statistical Modelling and Uncertainty Reporting

“The statistical section briefly mentions associations but does not detail: model type (e.g., logistic regression, GLM, Poisson, time-series), covariates included, handling of missing data, validation approach. Consider explicitly reporting: effect estimates with 95% confidence intervals, model diagnostics, and sensitivity checks (e.g., excluding borderline GDACS events). These additions would substantially improve interpretability.”

Response:

We appreciate the reviewer’s focus on statistical transparency and uncertainty reporting. In the current version of the manuscript, we already provide detailed information on model type, covariates, missing data handling, and internal validation, as well as effect estimates, confidence intervals, model diagnostics, and pre-specified sensitivity analyses. We summarise these elements and their locations here for clarity.

First, the model types used are explicitly described in Section 2.2 (Statistical analysis). All analyses are conducted at the composite event level. Associations between GDACS indicators and outcomes are quantified using Spearman’s rank correlation. For death counts, we use negative binomial regression to account for overdispersion, and results are reported as incidence-rate ratios (IRRs) with 95% confidence intervals. For field-hospital deployment (yes/no), we use a Firth penalised logistic regression, motivated by the limited number of deployment events and near-separation; this is introduced in the primary objectives and in the description of the analytical environment, and summarised in the model diagnostics table (Supplementary Table S2, Panel A).

Second, the covariates available and how they enter the models are detailed in Section 2.4 (Variables and Linkage) and summarised in the new Table 1 (“GDACS indicators and operational definitions used in this study”). The main predictors are the continuous GDACS score, alert level (Orange/Red), moment magnitude, hypocentre depth, and, in a subset, GDACS-modelled population exposed within 100 km. For rate-based analyses of deaths, exposure enters as a denominator (or offset), as specified in the Statistical Analysis section and co-primary endpoint description. We do not fit highly parameterised multivariable models beyond this parsimonious set, given the modest sample size and event counts.

Third, handling of missing data is described under “Missing data” in Section 2.8 (Methodological Considerations – Bias and Assumptions). We use complete-case analysis per model, and variable-wise missingness counts are tabulated in Supplementary Table S2 (Panel B). As noted in the Discussion, the amount of missingness is modest and was judged insufficient to warrant multiple imputation given the limited sample size and the primarily exploratory nature of the deployment model.

Fourth, internal validation and model diagnostics are reported in both the Methods and Supplementary material. For the field-hospital deployment model, discrimination (area under the receiver-operating characteristic curve, AUC), overall accuracy (Brier score), and calibration slope/intercept are specified in the primary objective and fully reported in Supplementary Table S2 (Panel A). In the Results, we highlight that the model achieves excellent discrimination (AUC ≈ 0.98) with acceptable calibration, acknowledging that this reflects internal performance only and does not constitute external validation. For the death models, we report IRRs with 95% confidence intervals and evaluate model fit via overdispersion checks and residual inspection.

Fifth, effect estimates with 95% confidence intervals are already presented for the regression models. For death counts, IRRs and their 95% confidence intervals are reported in the regression table and referenced in the Results. For the deployment model, we focus on predictive performance metrics (AUC, Brier score, calibration) rather than on the magnitude of the odds ratio, because the primary inferential interest is in predictive utility rather than in a fully adjusted causal effect estimate. Nonetheless, all model coefficients with standard errors and confidence intervals are available in the shared code and output.

Finally, sensitivity analyses are pre-specified and reported in Supplementary Table S1. These include (i) analyses using raw (non-composite) GDACS events, (ii) an alternative definition of deployment (broader temporary medical facilities), and (iii) alternative composite windows (±12/±48 hours; 100–200 km). Together, these robustness checks address concerns that the main associations might be driven by specific clustering choices or by the stricter deployment definition. We did not add an additional, ad hoc sensitivity analysis excluding “borderline GDACS events”, because this would further reduce effective sample size and overlap conceptually with the existing pre-specified robustness checks.

In view of the above, we did not modify the underlying models, but we have ensured that the Statistical Analysis, Variables and Linkage sections, and the new Table 1, as well as Supplementary Tables S1–S2, clearly document model types, covariates, missing data handling, internal validation metrics, effect estimates, and robustness analyses. We hope this summary clarifies the analytical structure and addresses the reviewer’s concern about interpretability.

5. Reviewer 2, comment 5 – Data Sources and Spatial Resolution

 “The phrase ‘subnational exposure and rapid damage assessment’ requires: clear naming of data sources, geographic resolution (district/province/municipality), and whether exposure data were updated dynamically or based on historical baselines. The manuscript would benefit from clarifying how spatial mismatches were managed if outcomes were recorded at a different administrative level.”

Response:

We thank the reviewer for this comment. The current analysis does not use any subnational administrative exposure datasets or rapid damage assessment products. All predictors and outcomes are defined at the level of composite earthquake events. Specifically, as detailed in Sections 2.3–2.4 and summarised in Table 1, the GDACS indicators used in the models are the continuous GDACS score, alert level (Orange/Red), moment magnitude, hypocentre depth, and, in a subset of events, the GDACS-modelled population exposed within 100 km of the epicentre. Outcomes are limited to field-hospital deployment (yes/no), total reported deaths, and deaths per 100,000 exposed where an explicit exposure estimate is available. These quantities are linked to each composite event rather than to district-, province-, or municipality-level administrative units, and we therefore do not perform spatial joins across mismatched administrative levels.

The phrase “subnational exposure and rapid damage assessment” appears in the Abstract, Discussion, and Conclusion solely to indicate potential future directions—namely, that integrating GDACS hazard signals with additional subnational exposure maps, damage assessments (e.g., remote-sensing products), and health-system status indicators could, in principle, improve the alignment between alerts and expected clinical burden in subsequent work. These data layers were not available in a globally consistent form for the 2020–2024 earthquake cohort analysed here and were not used in the present models. For this reason, there are no additional data sources, geographic resolutions, or spatial mismatch procedur

---

## [Editor Report · Decision Letter 1]

1 Dec 2025

Using GDACS to Anticipate Clinical and Operational Burden after Earthquakes: A Global Event-Level Analysis (2020–2024)

PONE-D-25-49086R1

Dear Dr. Aykut,

We’re pleased to inform you that your manuscript has been judged scientifically suitable for publication and will be formally accepted for publication once it meets all outstanding technical requirements.

Kind regards,

Anna Bernasconi, PhD

Academic Editor

PLOS ONE

Additional Editor Comments (optional):

Dear authors, the revised manuscript now meets PLOS One quality standards. We appreciate the improvements made with respect to the original version. The paper can be accepted in its current form.
---

## [Editor Report · Acceptance letter]

PONE-D-25-49086R1

PLOS One

Dear Dr. Aykut,

I'm pleased to inform you that your manuscript has been deemed suitable for publication in PLOS One. Congratulations! Your manuscript is now being handed over to our production team.

Kind regards,

on behalf of

Dr. Anna Bernasconi

Academic Editor

PLOS One